# Endoscopic Surgical Approach for a Mesiodens in the Nasal Cavity: A Rare Phenomenon

**DOI:** 10.3390/reports7020046

**Published:** 2024-06-13

**Authors:** Enzo Iacomino, Chiara Fratini, Federica Zoccali, Francesca Cambria, Matteo Laudani, Alberto Eibenstein, Christian Barbato, Marco de Vincentiis, Antonio Minni

**Affiliations:** 1Department of Life, Health, and Environmental Sciences, Università degli Studi di L’Aquila, 67100 L’Aquila, Italy; enzoiacomino74@hotmail.com; 2Department of Sense Organs, Sapienza University of Rome, Viale del Policlinico 155, 00161 Rome, Italy; chiara.fratini135@gmail.com (C.F.); federica.zoccali@uniroma1.it (F.Z.); francesca.cambria@uniroma1.it (F.C.); matteo.laudani@uniroma1.it (M.L.); marco.devincentiis@uniroma1.it (M.d.V.); 3Department of Applied Clinical and Biotechnological Sciences, University of Aquila, 67100 L’Aquila, Italy; alberto.eibenstein@univaq.it; 4Institute of Biochemistry and Cell Biology (IBBC-CNR), Sapienza University Rome, Policlinico Umberto I, 00161 Rome, Italy; 5Division of Otolaryngology-Head and Neck Surgery, ASL Rieti-Sapienza University, Ospedale San Camillo de Lellis, 02100 Rieti, Italy

**Keywords:** mesiodens, nasal septum, nasal fossa, surgical removal, endoscopic nasal surgical approach

## Abstract

The nasal cavity is a sporadic site for mesiodens, and if it is impacted in the lower nasal floor or localized in the nasal septum, it may cause various nasal symptoms such as nasal obstruction, recurrent rhinitis, and epistaxis. Early diagnosis is made through clinical findings and a cone-beam computed tomography (CBCT) scan, but a definite treatment plan has not yet been developed. This study aims to present a case of a mesiodens in a 27-year-old male, located in the nasal septum, an unusual and rare site, and its surgical removal using an endoscopic nasal approach with subperiosteal intranasal dissection. The result of the study appears significant because this technique led to fewer postoperative complications, and it appears to be safer and more effective than the traditional palatal or transoral approach. Moreover, the nasal endoscopic approach is more natural to the ear, nose, and throat (ENT) surgeons than the transoral approach.

## 1. Introduction

The mesiodens is a supernumerary tooth that derives its name from its location between the maxillary central incisors. In the literature, the prevalence of the mesiodens was reported to be between 0.15 and 1.9% [1]. They account for 80% of all supernumerary teeth and are approximately twice as frequent in men as in women and, most frequently, patients present a single supernumerary tooth [2,3]. The etiology of mesiodentes is not fully understood, and it could be considered a debated topic because a few factors play a key role in its occurrence, including developmental disturbances, inflammatory reactions, genetic predisposition, and trauma [4,5]. Several theories exist regarding mesiodentes, such as genetic susceptibility and environmental or hereditary factors. 

The increase of dental lamina could lead to the formation of extra teeth. Mesiodentes are more common in family members, suggesting heredity as an etiological factor, and not following a Mendelian pattern.

Ectopic eruption of a supernumerary tooth occurs in multiple locations, such as the maxillary sinus, mandibular condyle, coronoid process, orbits, and palate, through the skin and nasal cavity. The nasal cavity is a rarely reported site of a mesiodens’ location, ranging from 0.1% to 1% in the general population [6,7]. 

Some studies have reported that mesiodentes on the palatal side vary between 61.9% and 89%. Nazif et al. reported that 6% are in a labial position, 80% palatally, and the remaining 14% are located between the roots of the central incisors [8]. Mossaz et al. observed that 20.5% of the mesiodentes are in contact with the cortical bone of the nasal floor and 49% are about the nasopalatine canal [9].

The mesiodens can be asymptomatic for a considerable amount of time but may cause a variety of signs and symptoms such as mild facial pain, epistaxis, nasal obstruction, and rhinorrhea. They frequently interfere with the eruption of the permanent central incisor, causing displacement, rotation, and lack of eruption [10].

To determine the position of the mesiodens, a CBCT scan in axial and coronal views are used, and the treatment is surgical removal because although many patients remain asymptomatic, identifying and removing mesiodentes have been proven to be beneficial since a mesiodens has the potential to cause morbidity [11,12]. They should be removed to alleviate the symptoms and prevent complications such as caseous rhinitis with septal perforation, aspergillosis, and naso-oral fistula, and such surgery may be aided by real-time imaging such as fluoroscopy [13].

Previous studies have made classifications of the mesiodens according to its characteristics, including the location, crown direction, and morphology [14]; however, the correlation between classifications and surgical approaches have not been well discussed. Traditionally, mesiodentes are extracted via intraoral approaches, such as the transpalatal approach made by an incision in the palatal gingival sulcus, or through a vestibular or labial approach made by an incision above the edge of the gingiva on the labial side of the operative area, or both. The main advantage of the traditional labial approach is that it provides an excellent surgical view, but an osteotomy in the anterior maxillary area is likely to exacerbate postoperative swelling and pain. The palatal approach shows reduced aesthetic problems in the anterior maxillary area, and in addition, the degree of postoperative swelling is mild. However, an extensive osteotomy may reduce adjacent teeth’ blood supply and pulp vitality [15,16]. The vestibular approach with subperiosteal intranasal dissection as described by Sammartino G. et al. is a surgical technique that has not been intensively studied and a limited number of cases have been reported because some stomatologists are worried about damaging the nasal floor structure, breaking through the nasal mucosa into the trachea, or even causing airway obstruction by the misplacing the mesiodens [17].

We present a rare case of a mesiodens involving an unusual and rare site in a 27-year-old male, and its endoscopy management. The patient underwent surgical excision via an intranasal endoscopic approach. This technique demonstrates that the use of endoscopy to treat a Mesiodens located in the nasal septum provides a lower risk of complications and less postoperative morbidity than traditional approaches.

## 2. Detailed Case Presentation

In September 2022, a 27-year-old male was referred to the ENT Unit of San Salvatore Hospital (L’Aquila) for nasal blood-stained discharge, pain, and a blocked nasal passage. The medical history was essentially negative, the patient had no significant chronic conditions, he did not take home any therapy, there was no previous history of maxillofacial trauma or surgery, he had not visited the dentist for at least 3 years, and had never undergone an ear, nose, and throat examination (otorhinolaryngological evaluation).

The patient showed a nasal obstruction, recurrent rhinitis, epistaxis, and hyposmia for about 6 years, occasionally treated with local therapy consisting of corticosteroids and antihistamines without any improvement.

At the time of the ENT visit, the patient was not taking any medication, thinking that this nasal obstruction was due to an allergy. On examination with a nasal speculum, an inverted supernumerary tooth could not be visualized in the nasal cavities, but the preliminary exam revealed significant anterior deviation to the left side of the nasal septum. The nasal mucosa was edematous and hyperemic. Both the inferior nasal turbinates were hypertrophic, and the left nasal inferior turbinate touching the nasal septum determined a complete obstruction of the left nasal cavity. His intraoral dentition was normal with no cleft palate or other congenital anomalies. 

A cone-beam CT scan of the paranasal sinuses was performed, with the examination revealing nasal septum deviation, hypertrophia of both nasal turbinates, clear paranasal sinus, and the presence of a radiopaque lesion near the nasal septum with an attenuation of the mass that was the same as that of the oral teeth. It was localized in the maxillary midline area, it was set in the nasal septum and caused significant displacement of it. It was not detected during the clinical assessment because it was embedded in the nasal mucosa. A CT axial view of the intranasal mesiodens revealed its penetration into the nasal cavity, specifically located in the nasal septum. A coronal scan obtained with the bone window setting shows the crown portion of the impacted mesiodens within the base of the nasal septum (Figure 1A,B).

## 3. Surgical Technique

In 2022, the mesiodens removal was performed through an endoscopic nasal approach. Access via the nostrils was performed with a vertical hemitransfixion incision through the right side of the membranous septum. Dissecting scissors and Cottle elevators were then used to develop a sub-mucoperichondrial plane and dissected posteriorly to reveal the quadrangular cartilage. The robust mucoperichondrial flap was elevated and raised through the same incision after a second flap on the contralateral side. On the floor of the nasal fossa, an endoscopic osteotomy with a microdebrider was performed and by a 0-degree Hopkins rod endoscope, the mesiodens were shown. Surrounding the tooth, there was chronic inflammatory tissue. A twisting movement was employed to remove the mesiodens; once the tooth was luxated, the mesiodens would be grasped and extracted through the nostrils (Figure 2) followed by bleeding control. The mucoperichondrial flaps were laid back into position against the septum and sutured in Vicryl 3/0. Nasal packing was placed in both nasal fossae (Merocel, Medtronic, Mystic, CT, USA) to support the septal mucoperichondrial flaps and minimize the risk of septal hematoma formation. The extracted mesiodens was 14 mm in length, had a short cone-shaped root with a dilaceration at the apex, and a crown with incisor and canine teeth characteristics. The patient was discharged on the same day. The postoperative healing was uneventful. The pack was removed on the fifth postoperative day without any complications and the postoperative care consisted of a five-day course of Amoxicillin antibiotics and nonsteroidal anti-inflammatory drugs. 

## 4. Discussion

An intranasal supernumerary tooth is a rare phenomenon, and an ectopic intranasal tooth embedded in the mucosa of the nasal septum is very rare [18,19]. Nasal mesiodentes are typically observed as a single unilateral tooth and have a vague clinical presentation, often leading to them becoming easily overlooked. Patients are usually asymptomatic but if the mesiodens affects the lower nasal floor or is in the nasal septum, this may cause symptoms such as nasal obstruction, epistaxis, facial pain, and complications. Therefore, the diagnosis of intranasal teeth is primarily made according to the clinical and radiographic findings, with CBCT scanning being particularly helpful in confirming the diagnosis and planning treatment. Treatment of intranasal teeth typically involves early surgical extraction to relieve symptoms and prevent possible morbidity [20]. While various approaches exist to remove a nasal mesiodens, a definitive treatment plan has not yet been established.

There are different surgical approaches available, including the following: some surgeons prefer a palatal approach, associated with poor accessibility and the risk of neurovascular injury, such as damage to the nasopalatine nerve; a labial approach, despite the high risk of injury to the near-permanent teeth which requires excessive osteotomy; or the nasal floor approach, used for inverted mesiodentes but reported in a limited number of cases [21]. These techniques are accompanied by complications such as traumatizing or injuring the nasal mucosa or adjacent structures, which are the roots of the adjacent permanent teeth; therefore, to date, there is no generally accepted surgical procedure or a definite treatment plan for mesiodentes and the treatment varies for each case [22].

We opted for an endoscopic nasal surgical approach, performed by anterior rhinoscopy under endoscopic guidance with debriding for clear visualization to minimize injury to nearby structures. The access through the nostrils prevents the removal of palatal or buccal bone before accessing the mesiodens, and endoscopy can provide a direct view of the surgical site, good illumination, and a precise dissection to remove the nasal mesiodens. It improved accessibility to remote areas and magnification while allowing limited intranasal incision and elevation of small flaps, without compromising adequate exposure of the surgical site and the buccal or palatal area [23].

Endoscopic techniques can be challenging because of the frequent staining of the endoscope lens by blood from the incision site and difficulty in navigating the nasal passages. It may also be technically more challenging than traditional methods, and for this reason, both a high level of experience and good surgical skills of the operator are required [23].

This approach allows for a type of surgical procedure that, although usually employed by ENT specialists for septum correction or paranasal sinus surgery, in our experience proves to be effective in the removal of a mesiodens located in the nasal fossa, thanks to the use of endoscopy and an exclusively intranasal approach, which allowed for rapid and precise execution, complete removal of the supernumerary tooth, a reduction in complications related to vestibular or palatal incisions, and a reduced need for osteotomies with subsequent pain for the patient, along with minor problems such as lesions of the nasopalatine bundle or paresthesias.

It can be managed as a day-care procedure with reduced postoperative morbidity; in fact, in our case, no intraoperative or postoperative complications occurred, and our patient did not refer any symptoms or pain after the surgical procedure.

## 5. Conclusions

In conclusion, this case demonstrated that intranasal teeth are a rare form of ectopic teeth encountered in otolaryngology clinics that may cause a variety of nasal symptoms associated with a long story of misdiagnosis, and that early diagnosis and treatment are important to avoid their complications. The CT exam was essential to determine the tooth’s position and to help in the surgical planning. We used an endoscopic nasal approach with subperiosteal intranasal dissection, a useful approach for the exposure and removal of teeth impacted in the septum and localized in the floor of the nasal cavity.

This endoscopic technique appears more natural to the ENT surgeons than the transoral approaches, and the advantage seems that it could be safer and more effective than the traditional approaches, allowing for the lower risk of complications and postoperative morbidity, with no intraoperative or postoperative complications. 

## Figures and Tables

**Figure 1 reports-07-00046-f001:**
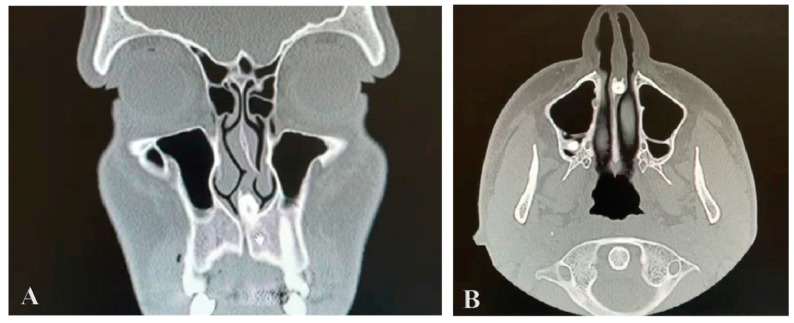
Computed tomography cut bone window coronal (**A**) and axial (**B**) views, showing the nasal tooth.

**Figure 2 reports-07-00046-f002:**
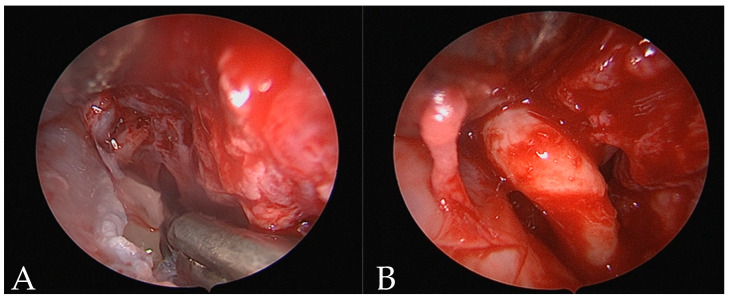
Endoscopic nasal view (**A**,**B**).

## Data Availability

The original contributions presented in the study are included in the article material, further inquiries can be directed to the corresponding author.

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
