# Peer review of "Endoscopic Surgical Approach for a Mesiodens in the Nasal Cavity: A Rare Phenomenon"

_reports, 2024, doi:10.3390/reports7020046_

Round 1
Reviewer 1 Report
Comments and Suggestions for Authors
This is a case reporting using endoscopic surgical approach for a Mesiodens in the nasal cavity. The case is interesting to clinicians working in relevant field.
Abstract
The background information is too much and should be substantially trimmed down to one to max two sentences. After reporting the study aim, there should be a good summary of the reported case and management reported in the abstract. I suggest to conclude with a significance of reporting this case study.
Introduction
Prevalence of Mesodens should be included with a search and reference to epidemiological studies. Tidy up the management strategies and discuss the merits and limitations of convention treatment: this provide a gap for reporting this case report.
Details Case presentation
Sex?
Social history and familial medical background can be reported. The suffering period should be reported, and what alleviative care delivered can be added.
Treatment /management should be presented as a separated section with discussing of the treatment and management of post-operative care.
Discussion
A comparison of the merits and limitations of different (convemntional and the preposed) diagnosis and management of mesiodens should be discussed.
Conclusion
Simplified the conclusion with a summary on the clinical significance of this case report.
Acknowledgement
Should acknowledge the patient for his/her consent for this report.
Comments on the Quality of English LanguageNeed copy editing
Author Response
June 5, 2024
Dear Editors and Reviewers:
Re: Revised version of the manuscript Reports-3054224
We express our gratitude to the editors and three reviewers for their valuable and insightful feedback on our original submission. We have carefully considered each comment and suggestion, and as a result, we have made significant revisions to our article to address the reviewers' concerns. We have indicated in red the modifications made in the revised version, addressing the reviewers' specific comments. Additionally, we have included some new information in the revised manuscript (also indicated in red). We hope that these changes have improved the manuscript and that the reviewers will find it suitable for publication in the Reports journal.
Reviewer #1
Comments and Suggestions for Authors
This is a case reporting using endoscopic surgical approach for a Mesiodens in the nasal cavity. The case is interesting to clinicians working in relevant field.
R1) Abstract
The background information is too much and should be substantially trimmed down to one to max two sentences. After reporting the study aim, there should be a good summary of the reported case and management reported in the abstract. I suggest to conclude with a significance of reporting this case study.
R1) The abstract was rewritten as suggested and we added line 23-29: “This study aims to present a case of mesiodens in a 27-year-old male, located in the nasal septum, an unusual and rare site, and its surgical removal using an endoscopic nasal approach with subperiosteal intranasal dissection. The result of the study appears significant because this technique led to fewer postoperative complications, and it appears to be safer and more effective than the traditional palatal or transoral approach. Moreover, the nasal endoscopic approach is more natural to the ear nose, and throat (ENT) surgeons than the transoral approach.”
R2) Introduction
Prevalence of Mesodens should be included with a search and reference to epidemiological studies. Tidy up the management strategies and discuss the merits and limitations of convention treatment: this provide a gap for reporting this case report.
R2) In the introduction, we added the following sentences:
Lines 30-44: In the literature, the prevalence of the mesiodens was reported between 0.15 - 1.9% [1]. They account for 80% of all supernumerary teeth and are approximately twice as frequent in men as in women and most frequently, patients present a single supernumerary tooth [2,3]. The etiology of mesiodens is not fully understood, and it could be considered a debated topic because a few factors play a key role in its occurrence: developmental disturbances, inflammatory reactions, genetic predisposition, and trauma [4,5]. Several theories exist regarding mesiodens, such as genetic susceptibility and environmental or hereditary factors. The activity of the dental lamina leading to the formation of the extra teeth, could increase. Mesiodens are more common in family members suggesting heredity as an etiological factor, and not following a Mendelian pattern.
R3) Details Case presentation
Sex? Social history and familial medical background can be reported. The suffering period should be reported, and what alleviative care delivered can be added.
The following sentences were added:
R3) Line 87-95: a 27-year-old male was referred to the ENT Unit of San Salvatore Hospital (L’Aquila) for nasal blood-stained discharge, pain, and blocked nasal passage. The medical history was essentially negative, the patient has no significant chronic conditions, he does not take-home therapy, there was no previous history of maxillofacial trauma or surgery and he had not visited the dentist for at least 3 years, and had never undergone an ear, nose, and throat examination (otorhinolaryngological evaluation). The patient referred nasal obstruction, recurrent rhinitis, epistaxis, and hyposmia for about 6 years, treated sometimes with local therapy consisting of corticosteroids and antihistamines without any improvement.
R4) Treatment /management should be presented as a separated section with discussing of the treatment and management of post-operative care.
R4) LINES 138-141: “The postoperative healing was uneventful. The pack was removed on the 5th postoperative day without any complications and the postoperative care consisted of a 5-day course of Amoxicillin antibiotics and nonsteroidal anti-inflammatory drugs.”
R5) Discussion
A comparison of the merits and limitations of different (convemntional and the preposed) diagnosis and management of mesiodens should be discussed.
R5) In the manuscript, we added the following sentences:
Lines 170-192: “These techniques are accompanied by complications such as traumatizing or injuring the nasal mucosa or adjacent structures, that is the roots of the adjacent permanent teeth so, to date, there is no generally accepted surgical procedure or a definite treatment plan for mesiodens and treatment varies for each case [22].
We opted for an endoscopic nasal surgical approach, performed by anterior rhinoscope under endoscopic guidance with debriding for clear visualization to minimize injury to nearby structures. The access through the nostrils prevents the removal of palatal or buccal bone before accessing the mesiodens and endoscopy can provide a direct view of the surgical site, a good illumination, and a precise dissection to remove nasal mesiodens. It improved accessibility to remote areas and magnification while allowing limited intranasal incision and elevation of small flaps without compromising adequate exposure of the surgical site and the buccal or palatal area [23]. Endoscopic techniques can be challenging because of frequent staining of the endoscope lens by blood from the incision site and difficulty in navigating the nasal passages it may be technically more challenging than traditional methods, and for this reason, both a high level of experience and good surgical skills of the operator is required [24].
This approach allows for a type of surgical procedure that, although usually employed by ENT specialists for septum correction or paranasal sinus surgery, in our experience proves to be effective in the removal of mesiodens located in the nasal fossa thanks to the use of endoscopy and an exclusively intranasal approach, it allows for rapid and precise execution, complete removal of the supernumerary tooth, a reduction in complications related to vestibular or palatal incisions, and a reduced need for osteotomies with subsequent pain for the patient and potential issues such as injury to the nasopalatine bundle, or paresthesia.
It can be managed as a day-care procedure with reduced postoperative morbidity, in fact in our case no intraoperative or postoperative complications occurred and our patient didn’t refer any symptoms or pain after the surgical procedure.”
R6) Conclusion
Simplified the conclusion with a summary on the clinical significance of this case report.
R6) The conclusion section was entirely rewritten: Lines 195-206: “In conclusion, this case demonstrated that intranasal teeth are a rare form of ectopic teeth encountered in otolaryngology clinics, that may cause a variety of nasal symptoms associated with a long story of misdiagnosis and that early diagnosis and treatment are important to avoid their complications. CT exam was essential to determine the tooth’s position and to help in the surgical planning. We use an endoscopic nasal approach with subperiosteal intranasal dissection, a useful approach for the exposure and removal of teeth impacted in the septum and localized in the floor of the nasal cavity.This endoscopic technique appears more natural to the Ent surgeons than the transoral approaches and the advantage seems that it could be safer and more effective than the traditional approaches for the lower risk of complications and postoperative morbidity no intraoperative or postoperative complications occurred.”
R7) Acknowledgement
Should acknowledge the patient for his/her consent for this report.
R7) Lines 219-220. Acknowledgments: Thank you to the patients who participated in the study and for consenting to the publication of this report.
The authors would like to express their gratitude to Reviewer 1 for the insightful feedback. Your feedback on the communication of these potentially significant findings will contribute to enhancing the clarity and impact of our research.
Sincerely
Dr. Christian Barbato and Prof. Antonio Minni
Reviewer 2 Report
Comments and Suggestions for Authors
The manuscript by Iacomino et al. presents a rare clinical case of mesiodens located in the nasal cavity and its endoscopic removal. The paper can be interesting to the scientific society. However, some corrections and modifications are necessary before its publishing.
The information in the Introduction, Discussion, and Conclusion sections is similar and repetitive.
As a simple suggestion, the Introduction can focus on the etiology, pathogenesis, prevalence, and clinical findings.
The case report is well-described. Antibiotic and anti-inflammatory drugs can be specified using generic names.
The diagnosis and treatment options can be described in the Discussion section, using appropriate references.
"The formation of mesiodens is a debated topic, several theories exist..." (lines 118-119) - these theories should be listed or described.
"There are different surgical approaches: some surgeons prefer a palatal approach, which is widely used for surgical extraction, but it may be associated with poor accessibility and a risk of neurovascular injury such as damage to the nasopalatine nerve. An alternative approach is the labial one despite the high risk of injury to near-permanent teeth which requires excessive osteotomy. Otherwise, the nasal floor approach is used for inverted mesiodens." (lines 132-137) - appropriate references should be added. The indications and limitations of the above-mentioned approaches can be discussed.
The Conclusion section should summarize the information from the previous sections. In this case, it looks like a repetition of both Introduction and Discussion. The statements in this section are not supported by relevant references in the Discussion.
Comments on the Quality of English LanguageEnglish is understandable. However, some major corrections are required.
Author Response
June 5, 2024
Dear Editors and Reviewers:
Re: Revised version of the manuscript Reports-3054224
We express our gratitude to the editors and three reviewers for their valuable and insightful feedback on our original submission. We have carefully considered each comment and suggestion, and as a result, we have made significant revisions to our article to address the reviewers' concerns. We have indicated in red the modifications made in the revised version, addressing the reviewers' specific comments. Additionally, we have included some new information in the revised manuscript (also indicated in red). We hope that these changes have improved the manuscript and that the reviewers will find it suitable for publication in the Diseases journal.
Reviewer #2
Comments and Suggestions for Authors
The manuscript by Iacomino et al. presents a rare clinical case of mesiodens located in the nasal cavity and its endoscopic removal. The paper can be interesting to the scientific society. However, some corrections and modifications are necessary before its publishing.
R1) The information in the Introduction, Discussion, and Conclusion sections is similar and repetitive.
R1) The Introduction, Discussion, and Conclusion sections were reorganized and several parts were discussed adding new references.
R2) As a simple suggestion, the Introduction can focus on the etiology, pathogenesis, prevalence, and clinical findings.
R2) Lines 30-84: new Introduction section:
“Mesiodens is a supernumerary tooth that derives its name from its location between the maxillary central incisors. In the literature, the prevalence of the mesiodens was reported between 0.15 - 1.9% [1]. They account for 80% of all supernumerary teeth and are approximately twice as frequent in men as in women and most frequently, patients present a single supernumerary tooth [2,3]. The etiology of mesiodens is not fully understood, and it could be considered a debated topic because a few factors play a key role in its occurrence: developmental disturbances, inflammatory reactions, genetic predisposition, and trauma [4,5]. Several theories exist regarding mesiodens, such as genetic susceptibility and environmental or hereditary factors. The activity of the dental lamina leading to the formation of the extra teeth, could increase. Mesiodens are more common in family members suggesting heredity as an etiological factor, and not following a Mendelian pattern.Ectopic eruption of supernumerary tooth occurs in multiple locations such as the maxillary sinus, mandibular condyle, coronoid process, orbits, and palate, through the skin and nasal cavity. The nasal cavity is a rare site of mesiodens location reported ranging from 0.1% to 1% in the general population (6,7). Some studies have reported that mesiodens on the palatal side vary between 61.9% and 89%. Nazif et al. reported that 6% are in a labial position, 80% palatally, and the remaining 14% are located between the roots of the central incisors [8]. Mossaz et al. observed that 20.5% of the mesiodens are in contact with the cortical bone of the nasal floor and 49% about the nasopalatine canal [9].Mesiodens can be asymptomatic for a considerable amount of time but may cause a variety of signs and symptoms such as mild facial pain, epistaxis, nasal obstruction, and rhinorrhea. They frequently interfere with the eruption of the permanent central incisor causing displacement, rotation, and lack of eruption [10].To determine the position of the mesiodens a CBCT scan in axial and coronal views is used, and the treatment is surgical mesiodens removal because although many patients remain asymptomatic, identifying and removing mesiodens have been proven to be beneficial since mesiodens have the potential to cause morbidity [11,12]. They should be removed to alleviate the symptoms and prevent complications such as rhinitis caseosa with septal perforation, aspergillosis, and naso-oral fistula and surgery may be aided by real-time imaging such as fluoroscopy [13].Previous studies made classifications of mesiodens according to the characteristics of mesiodens, including the location, crown direction, and morphology [14] and the correlation between classifications and surgical approaches had not been well discussed. Traditionally mesiodens are extracted via intraoral approaches such us transpalatal, made by an incision in the palatal gingival sulcus, or through vestibular or labial approach made by an incision above the edge of the gingiva on the labial side of the operative area, both. The main advantage of the traditional labial approach is to provide an excellent surgical view, but osteotomy in the anterior maxillary area is likely to exacerbate postoperative swelling and pain. The palatal approach showed reduced aesthetic problems in the anterior maxillary area, and in addition, the degree of postoperative swelling is mild, and an extensive osteotomy may reduce the blood supply and pulp vitality of adjacent teeth [15, 16]. The vestibular approach, with subperiosteal intranasal dissection, described by Sammartino G. et al., is a surgical technique that has not been intensively studied and a limited number of cases have been reported, because some stomatologists are worried about damaging the nasal floor structure, breaking through the nasal mucosa into the trachea, or even causing airway obstruction by the misplacing mesiodens [17]. We present a rare case of mesiodens involving unusual and rare sites in a 27-year-old male, and its endoscopy management. The patient underwent surgical excision via an intranasal endoscopic approach. This technique demonstrates that endoscopy to treat Mesiodens located in the nasal septum provides a lower risk of complications and less postoperative morbidity than traditional approaches.”
R3) The case report is well-described. Antibiotic and anti-inflammatory drugs can be specified using generic names.
R3) the requested information was added: Lines 135-138: The postoperative healing was uneventful. The pack was removed on the 5th postoperative day without any complications and the postoperative care consisted of a 5-day course of Amoxicillin antibiotics and nonsteroidal anti-inflammatory drugs.
R4) The diagnosis and treatment options can be described in the Discussion section, using appropriate references.
R4) The discussion section was reorganized and a new text was added Lines 167-191 and new references were included:
These techniques are accompanied by complications such as traumatizing or injuring the nasal mucosa or adjacent structures, that is the roots of the adjacent permanent teeth so, to date, there is no generally accepted surgical procedure or a definite treatment plan for mesiodens and treatment varies for each case [22].We opted for an endoscopic nasal surgical approach, performed by anterior rhinoscope under endoscopic guidance with debriding for clear visualization to minimize injury to nearby structures. The access through the nostrils prevents the removal of palatal or buccal bone before accessing the mesiodens and endoscopy can provide a direct view of the surgical site, a good illumination, and a precise dissection to remove nasal mesiodens. It improved accessibility to remote areas and magnification while allowing limited intranasal incision and elevation of small flaps without compromising adequate exposure of the surgical site and the buccal or palatal area [23].Endoscopic techniques can be challenging because of frequent staining of the endoscope lens by blood from the incision site and difficulty in navigating the nasal passages it may be technically more challenging than traditional methods, and for this reason, both a high level of experience and good surgical skills of the operator is required [24].This approach allows for a type of surgical procedure that, although usually employed by ENT specialists for septum correction or paranasal sinus surgery, in our experience proves to be effective in the removal of mesiodens located in the nasal fossa thanks to the use of endoscopy and an exclusively intranasal approach, it allows for rapid and precise execution, complete removal of the supernumerary tooth, a reduction in complications related to vestibular or palatal incisions, and a reduced need for osteotomies with subsequent pain for the patient and potential issues such as injury to the nasopalatine bundle, or paresthesia.It can be managed as a day-care procedure with reduced postoperative morbidity, in fact in our case no intraoperative or postoperative complications occurred and our patient didn’t refer any symptoms or pain after the surgical procedure.
R5) "The formation of mesiodens is a debated topic, several theories exist..." (lines 118-119) - these theories should be listed or described.
R5) In the introduction section these theories were described.
R6) "There are different surgical approaches: some surgeons prefer a palatal approach, which is widely used for surgical extraction, but it may be associated with poor accessibility and a risk of neurovascular injury such as damage to the nasopalatine nerve. An alternative approach is the labial one despite the high risk of injury to near-permanent teeth which requires excessive osteotomy. Otherwise, the nasal floor approach is used for inverted mesiodens." (lines 132-137) - appropriate references should be added. The indications and limitations of the above-mentioned approaches can be discussed.
R6) As reported above the discussion of different surgical approaches were discussed.
R7) The Conclusion section should summarize the information from the previous sections. In this case, it looks like a repetition of both Introduction and Discussion. The statements in this section are not supported by relevant references in the Discussion.
R7) The conclusion section was rewritten following the new discussion section as requested. Lines 193-203: “In conclusion, this case demonstrated that intranasal teeth are a rare form of ectopic teeth encountered in otolaryngology clinics, that may cause a variety of nasal symptoms associated with a long story of misdiagnosis and that early diagnosis and treatment are important to avoid their complications. CT exam was essential to determine the tooth’s position and to help in the surgical planning. We use an endoscopic nasal approach with subperiosteal intranasal dissection, a useful approach for the exposure and removal of teeth impacted in the septum and localized in the floor of the nasal cavity. This endoscopic technique appears more natural to the Ent surgeons than the transoral approaches and the advantage seems that it could be safer and more effective than the traditional approaches for the lower risk of complications and postoperative morbidity no intraoperative or postoperative complications occurred.
The authors would like to express their gratitude to Reviewer 2 for the insightful feedback. Your feedback on the communication of these potentially significant findings will contribute to enhancing the clarity and impact of our research.
Sincerely
Dr. Christian Barbato and Prof. Antonio Minni
Reviewer 3 Report
Comments and Suggestions for Authors
1. Explain all abbreviations at first use, both in the abstract and in the main text of the manuscript.
2. State a short aim of the report, at the end of the abstract and at the end of the introduction.
3. Are there any other methods of removal of mesiodens been reported in the literature. If so, they should be elaborated briefly in the introduction.
4. In addition to using multiplanar radiographs, real time surgical imaging such as fluoroscopy could also be used. A suggested reference in this regard (Removal of an Orthodontic Mini-Screw Displaced into the Lateral Pharyngeal Space: A Case Report and Review of Pertinent Literature. Cureus. 2024 Jan 15;16(1):e52343. doi: 10.7759/cureus.52343.)
5. A short literature review about intranasal or ectopic mesiodens and their reported removal techniques would benefit the readers. (Please see above suggested reference regarding the same).
6. Discussion - please see above comment to add more details to the discussion. Also, add information about potential advantages of the endoscopic technique over conventional surgical removal. Similarly, included technical challenged and complications, which could be encountered with the endoscopic technique.
Minor comments/correction
- lines 49-50: To be corrected as "... CT scans in axial and coronal views are used ..."
- line 71: To be corrected as "... inverted supernumerary tooth ..."
- line 73: To be corrected as "... edematous and hyperemic ..."
- line 79: To be corrected as "... hypertrophy of both nasal turbinates ..."
Comments on the Quality of English Language
Please check the entire manuscript for English language, grammar, spelling and word use.
Author Response
June 5, 2024
Dear Editors and Reviewers:
Re: Revised version of the manuscript Reports-3054224
We express our gratitude to the editors and three reviewers for their valuable and insightful feedback on our original submission. We have carefully considered each comment and suggestion, and as a result, we have made significant revisions to our article to address the reviewers' concerns. We have indicated in red the modifications made in the revised version, addressing the reviewers' specific comments. Additionally, we have included some new information in the revised manuscript (also indicated in red). We hope that these changes have improved the manuscript and that the reviewers will find it suitable for publication in the Diseases journal.
Reviewer #3
Comments and Suggestions for Authors
- Explain all abbreviations at first use, both in the abstract and in the main text of the manuscript.
- All abbreviations were explained.
- State a short aim of the report, at the end of the abstract and at the end of the introduction.
- A short aim of the report was added in the abstract and introduction section as follow:
Lines 23-29: “This study aims to present a case of mesiodens in a 27-year-old male, located in the nasal septum, an unusual and rare site, and its surgical removal using an endoscopic nasal approach with subperiosteal intranasal dissection. The result of the study appears significant because this technique led to fewer postoperative complications, and it appears to be safer and more effective than the traditional palatal or transoral approach. Moreover, the nasal endoscopic approach is more natural to the ear nose, and throat (ENT) surgeons than the transoral approach.”
Lines 80-84: “We present a rare case of mesiodens involving unusual and rare sites in a 27-year-old male, and its endoscopy management. The patient underwent surgical excision via an intranasal endoscopic approach. This technique demonstrates that endoscopy to treat Mesiodens located in the nasal septum provides a lower risk of complications and less postoperative morbidity than traditional approaches.”
- Are there any other methods of removal of mesiodens been reported in the literature. If so, they should be elaborated briefly in the introduction.
- Methods of mesiodens removal were described in the Introduction section as requested by Reviewers 2 and 3 as reported in the manuscript text.
Introduction lines 65-79: “Previous studies made classifications of mesiodens according to the characteristics of mesiodens, including the location, crown direction, and morphology [14] and the correlation between classifications and surgical approaches had not been well discussed. Traditionally mesiodens are extracted via intraoral approaches such us transpalatal, made by an incision in the palatal gingival sulcus, or through vestibular or labial approach made by an incision above the edge of the gingiva on the labial side of the operative area, both. The main advantage of the traditional labial approach is to provide an excellent surgical view, but osteotomy in the anterior maxillary area is likely to exacerbate postoperative swelling and pain. The palatal approach showed reduced aesthetic problems in the anterior maxillary area, and in addition, the degree of postoperative swelling is mild, and an extensive osteotomy may reduce the blood supply and pulp vitality of adjacent teeth [15, 16]. The vestibular approach, with subperiosteal intranasal dissection, described by Sammartino G. et al., is a surgical technique that has not been intensively studied and a limited number of cases have been reported, because some stomatologists are worried about damaging the nasal floor structure, breaking through the nasal mucosa into the trachea, or even causing airway obstruction by the misplacing mesiodens [17].
- In addition to using multiplanar radiographs, real time surgical imaging such as fluoroscopy could also be used. A suggested reference in this regard (Removal of an Orthodontic Mini-Screw Displaced into the Lateral Pharyngeal Space: A Case Report and Review of Pertinent Literature. Cureus. 2024 Jan 15;16(1):e52343. doi: 10.7759/cureus.52343.)
- The reference was added to the text.
- 5. A short literature review about intranasal or ectopic mesiodens and their reported removal techniques would benefit the readers. (Please see above suggested reference regarding the same).
- Many references were added or removed as suggested.
- Khambete N, Kumar R. Genetics and presence of non-syndromic supernumerary teeth: A mystery case report and review of literature. Contemp Clin Dent. 2012 Oct;3(4):499-502. doi: 10.4103/0976-237X.107455.
- Kantaputra PN, Tripuwabhrut K, Anthonappa RP, Chintakanon K, Ngamphiw C, Adisornkanj P, Porntrakulseree N, Olsen B, Intachai W, Hennekam RC, Vieira AR, Tongsima S. Heterozygous Variants in FREM2Are Associated with Mesiodens, Supernumerary Teeth, Oral Exostoses, and Odontomas. Diagnostics (Basel). 2023 Mar 23;13(7):1214. doi: 10.3390/diagnostics13071214.
- Aoun G, Nasseh I. Mesiodens Within the Nasopalatine Canal: An Exceptional Entity. Clin Pract. 2016 Dec 7;6(4):903. doi: 10.4081/cp.2016.903.
- Removal of an Orthodontic Mini-Screw Displaced into the Lateral Pharyngeal Space: A Case Report and Review of Pertinent Literature. Cureus. 2024 Jan 15;16(1):e52343. doi: 10.7759/cureus.52343
- Giancotti A, Grazzini F, De Dominicis F, Romanini G, Arcuri C. Multidisciplinary evaluation and clinical management of mesiodens. J Clin Pediatr Dent. 2002 Spring;26(3):233-7.
- Chen A, Huang JK, Cheng SJ, Sheu CY. Nasal teeth: report of three cases. AJNR Am J Neuroradiol. 2002 Apr;23(4):671-3.
- Discussion - please see above comment to add more details to the discussion. Also, add information about potential advantages of the endoscopic technique over conventional surgical removal. Similarly, included technical challenged and complications, which could be encountered with the endoscopic technique.
- The Discussion section was reorganized and many sentences were added as follow:
Lines167-191: “These techniques are accompanied by complications such as traumatizing or injuring the nasal mucosa or adjacent structures, that is the roots of the adjacent permanent teeth so, to date, there is no generally accepted surgical procedure or a definite treatment plan for mesiodens and treatment varies for each case [22].We opted for an endoscopic nasal surgical approach, performed by anterior rhinoscope under endoscopic guidance with debriding for clear visualization to minimize injury to nearby structures. The access through the nostrils prevents the removal of palatal or buccal bone before accessing the mesiodens and endoscopy can provide a direct view of the surgical site, a good illumination, and a precise dissection to remove nasal mesiodens. It improved accessibility to remote areas and magnification while allowing limited intranasal incision and elevation of small flaps without compromising adequate exposure of the surgical site and the buccal or palatal area [23].Endoscopic techniques can be challenging because of frequent staining of the endoscope lens by blood from the incision site and difficulty in navigating the nasal passages it may be technically more challenging than traditional methods, and for this reason, both a high level of experience and good surgical skills of the operator is required [24].This approach allows for a type of surgical procedure that, although usually employed by ENT specialists for septum correction or paranasal sinus surgery, in our experience proves to be effective in the removal of mesiodens located in the nasal fossa thanks to the use of endoscopy and an exclusively intranasal approach, it allows for rapid and precise execution, complete removal of the supernumerary tooth, a reduction in complications related to vestibular or palatal incisions, and a reduced need for osteotomies with subsequent pain for the patient and potential issues such as injury to the nasopalatine bundle, or paresthesia.It can be managed as a day-care procedure with reduced postoperative morbidity, in fact in our case no intraoperative or postoperative complications occurred and our patient didn’t refer any symptoms or pain after the surgical procedure.”
Minor comments/correction
- lines 49-50: To be corrected as "... CT scans in axial and coronal views are used ..."
-lines 58-59: corrected
- line 71: To be corrected as "... inverted supernumerary tooth ..."
-lines 98: corrected
- line 73: To be corrected as "... edematous and hyperemic ..."
-lines 100: corrected
- line 79: To be corrected as "... hypertrophy of both nasal turbinates ..."
-lines 101: corrected
The authors value your constructive feedback, which will undoubtedly contribute to the refinement and future direction of our research efforts.
Sincerely
Dr. Christian Barbato and Prof. Antonio Minni
Round 2
Reviewer 1 Report
Comments and Suggestions for Authors
The authors have addressed my comments. I suggest professional language editing before publication.
Comments on the Quality of English Language
Suggest professional language editing.
Reviewer 2 Report
Comments and Suggestions for Authors
The manuscript is improved and can be published in the present form.
Comments on the Quality of English LanguageMinor editing of English is required.
Reviewer 3 Report
Comments and Suggestions for Authors
I appreciate the authors for their efforts in improving the manuscript by revising it based on the review comments.